

# Constraining low-frequency variability in climate projections to predict climate on decadal to multi-decadal time scales - a 'poor-man' initialized prediction system

Rashed Mahmood[1,2], Markus G. Donat[1,3], Pablo Ortega[1], Francisco J. Doblas-Reyes[1,3], Carlos Delgado-
Torres[1] , Margarida Samsó[1], Pierre-Antoine Bretonnière[1]

[1]Barcelona Supercomputing Center, Barcelona, Spain
[2]University of Montreal, Montreal, Canada
[3]Institució Catalana de Recerca i Estudis Avançats (ICREA), Barcelona, Spain

*Correspondence to*: Rashed Mahmood (rashed.mahmood@bsc.es)

**Abstract.** Near-term projections of climate change are subject to substantial uncertainty from internal climate variability. Here we present an approach to reduce this uncertainty by sub-selecting those ensemble members that more closely resemble observed patterns of ocean temperature variability immediately prior to a certain start date. This constraint aligns the observed and simulated variability phases and is conceptually similar to initialization in seasonal to decadal climate predictions. We apply this variability constraint to large multi-model projection ensembles from the Coupled Model Intercomparison Project phase 6 (CMIP6), consisting of more than 200 ensemble members, and evaluate the skill of the constrained ensemble in predicting the observed near-surface temperature, sea-level pressure and precipitation on decadal to multi-decadal time scales.

We find that the constrained projections show significant skill in predicting the climate of the following ten to twenty years, and added value over the ensemble of unconstrained projections. For the first decade after applying the constraint, the global patterns of skill are very similar and can even outperform those of the multi-model ensemble mean of initialized decadal hindcasts from the CMIP6 Decadal Climate Prediction Project (DCPP). In particular for temperature, larger areas show added skill in the constrained projections compared to DCPP, mainly in the Pacific and some neighboring land regions. Temperature and sea-level pressure in several regions are predictable multiple decades ahead, and show significant added value over the unconstrained projections for forecasting the first two decades and the 20-year averages. We further demonstrate the suitability of regional constraints to attribute predictability to certain ocean regions. On the example of global average temperature changes, we confirm the role of Pacific variability in modulating the reduced rate of global warming in the early 2000s, and demonstrate the predictability of reduced global warming rates over the following 15 years based on the climate conditions leading up to 1998. Our results illustrate that constraining internal variability can significantly improve the accuracy of near-term climate change estimates for the next few decades.



## 1 Introduction

In the context of ongoing climate change, predicting the climate evolution over the coming decades is important to enable targeted adaptation to the anticipated changes. While increasing greenhouse gas concentrations cause a general global warming (IPCC, 2021), different modes of climate variability can regionally amplify or counteract the warming-related

effects.

To obtain information about the expected climate in the future, climate projections simulate the responses of the Earth system to specified radiative forcing scenarios (Eyring et al., 2016; Taylor et al., 2012). These climate projections are affected by different uncertainties related to the forcing scenario chosen, the climate model used, and the phasing of internal variability. For projections of near-term climate change in the next 20 to 30 years, internal variability is the dominating

source of uncertainty at regional scales (Hawkins and Sutton, 2009, 2011; Lehner, 2020), while at longer time scales the scenario choice becomes increasingly important.

Decadal predictions, initialized towards observational states, are designed to exploit the predictability arising from both internal climate variability (Meehl et al., 2021, Kushnir et al., 2019), which is achieved by phasing in the variability modes in the model simulations with our best estimate of the variability of the real-world climate, and from externally forced

changes (Doblas-Reyes et al., 2013). These initialized decadal predictions show significant improvements in terms of added skill as compared to the uninitialized projections (e.g. Smith et al., 2019, 2020). However, the decadal predictions often suffer from initialization shocks and/or the subsequent drift towards the model's preferred climate state which can significantly reduce the overall skill of a decadal prediction system (e.g. Bilbao et al., 2021). In addition, the decadal predictions involve running very large ensembles of simulations and are therefore computationally expensive, which has

traditionally limited their production to the next ten years with relatively small ensemble sizes (Boer et al., 2016).

As an alternative, constraining decadal variability in large ensembles of future projection simulations can improve climate information and reduce uncertainty of projections for the next few decades. Different approaches have been explored to constrain internal variability in climate projections (e.g. Hegerl et al., 2021). An important advantage of these approaches, based on sub-selecting members of a set of transient climate simulations, is that predictions made based on these simulations

are consistent with the model-specific climate attractor, and not affected by shock, drift or related artifacts (Hazeleger et al., 2013; Smith et al., 2013; Bilbao et al., 2021). On seasonal to interannual timescales, Ding et al. (2018) made skilful predictions of tropical Pacific sea surface temperatures (SST) by finding model analogues similar to the observed state and using the subsequent trajectories of those analogues as forecasts. Similarly, Menary et al. (2021) developed an analogue approach to predict decadal-scale variations in the North Atlantic region.

On decadal to multi-decadal timescales, Befort et al (2020) and Mahmood et al (2021) have recently proposed approaches to constrain projections based on their agreement with decadal predictions and demonstrated some added value beyond the time period covered by decadal predictions. However, the aforementioned limitations affecting the initialized decadal predictions can also limit the added value of the constrained ensembles. Here we implement an approach similar to Mahmood et al.





(2021), however using climate observations for the constraining criteria instead of decadal prediction data. In particular, we
use multi-annual averages of sea-surface temperature (SST) anomaly patterns as constraining criterion. In essence, this
method exploits ensembles of already available model simulations to subselect those members in closest agreement with the
contemporary and/or immediately preceding observed SST anomaly patterns. Such member selection method thus works as
a 'poor man' initialisation to predict climate in the following decades.

In the following we describe the data and approach used to implement the 'poor man' initialized prediction system (Section
2). We then demonstrate its application and evaluate the skill in predicting temperature, sea level pressure and precipitation
globally, in comparison to state-of-the-art initialized predictions contributing to the Decadal Climate Prediction Project
(DCPP; Boer et al., 2016), and discuss the sensitivity to some of the various choices to be made during the selection
procedure. We further outline the applicability of this approach to attribute predictability of specific decadal-scale
phenomena to certain ocean regions (Section 3). We conclude this paper with a summary and discussion of this 'poor man'
initialized prediction approach in the context of other existing prediction systems (Section 4).

## 2 Data and Methods

We use climate model simulation data from the Coupled Model Intercomparison Project phase 6 (CMIP6) simulations
(Eyring et al., 2016). A total of 212 ensemble members from 32 different models were available, composed of  transient
historical simulations (hereafter referred to as "unconstrained", see Table S1 in supplementary information) until 2014 and
continued  with future projection simulations following the shared socioeconomic pathway (SSP2-45) forcing scenario (from
2015 onwards). We also use 93 members from 9 different models of the CMIP6/DCPP-A initialized decadal hindcasts in
order to evaluate the skill of the constrained projections in comparison to actual initialized predictions.

The observational data set used in this study to constrain the climate projections is the  Extended Reconstructed Sea Surface
Temperature version 5 dataset (Huang et al., 2017) from the National Oceanic and Atmospheric Administration (NOAA).
Surface temperature from HadCRUT4.6 (Morice et al., 2012), sea level pressure (SLP) from Japanese 55-year Reanalysis
(JRA-55; Kobayashi et al., 2015) and precipitation from Global Precipitation Climatology Center version-2018 (Schamm et
al., 2014) were used to evaluate the hindcasts on decadal and multi-decadal timescales. All data sets used in this study were
converted to monthly mean anomalies from the reference climatological period of 1981-2010. We also evaluated the skill of
the hindcasts using a different set of observational data for surface temperature, SLP and precipitation (see TextS1 in
supplementary material), to confirm the robustness of the results for different reference datasets.

The constraining procedure involves comparing SST anomaly patterns of individual unconstrained members with the
corresponding observed anomalies averaged over a given period that precedes the start of the prediction, by means of area-
weighted spatial pattern correlation. To do so, all SST datasets from both models and observations were regridded to a
common regular 3°x3° grid. For each start date, based on these anomaly pattern correlations, the unconstrained ensemble
members were ranked (Fig. 1) and the top ranking 30 members (referred to as "Best30") were chosen for forecasting up to
20 years after the initialization period. Since the choice of selecting 30 members is somewhat arbitrary, the sensitivity to





selecting a different number of members is further addressed in section 3.2. We use 9-year averages for most of the analyses, but additionally tested other averaging periods for the constraints to assess the sensitivity of the predictions to this parameter. Using 9-year averages, in order to start a constrained prediction from January 1961, the nine year mean SST

anomalies from January 1952 to December 1960 were used to select the Best30 members. This procedure was repeated every year and the Best30 ensembles were selected based on the SST anomaly comparisons of 1953-1961 (for predictions starting in 1962), 1954-1962 (for predictions starting in 1963), 1955-1963 (for predictions starting in 1964), and so on. While the constrained projections can be used to make climate predictions for as long as the projections are run, in this study we focus on the forecast periods of years 1-10, 11-20, and 1-20 after the 'initialization' (meaning selection of members closest

to the observational state). To evaluate the 20-year mean hindcasts against observational data sets, the final constraining period considered goes from January 1991 to December 1999 for predicting January 2000 till December 2019. Therefore a total of 40 start dates were used for the hincasts. For real time prediction purposes the constraint would use the most recent nine years.

As also discussed in Mahmood et al. (2021), the constraint involves a number of choices. For example, regional SST

anomalies can be used instead of using global SSTs to rank and subselect the Best30 ensemble. Similarly, as discussed above time periods covering different numbers of years (rather than using 9-year average SST anomalies) can also be used for determining the Best30 members most similar to observations. The sensitivity to these regional and temporal initializations and other potential choices are evaluated in sections 3.2 and 3.3.

The skill of the hindcasts is evaluated with two different deterministic metrics: the anomaly correlation coefficient (ACC) to

test the phase agreement between the climate model ensemble means (unconstrained, Best30 and DCPP) and observational data sets (Goddard et al., 2013), and the residual correlations to evaluate the added value of Best30 over the unconstrained ensemble mean after removing an estimate of the forced signal following Smith et al. (2019). The statistical significance of the ACC and residual correlation is estimated based on a two-tailed Student's t-test after taking into account the temporal autocorrelation (Guemas et al., 2014). The results are considered statistically significant when the null hypothesis of no

correlation can be rejected with $p<0.05$. We use the residual correlation to identify added value for surface temperature due to strong contribution of the forced signal in ACC for this variable, resulting in small ACC differences between the constrained/initialized and unconstrained ensembles. For precipitation and sea level pressure the forcing has smaller contributions to the skill. For those variables we instead use ACC differences between Best30 and unconstrained ensembles to illustrate the added value of the constrained ensemble. We however note that using residual correlations leads to very

similar results as the ACC differences (not shown). In order to test the statistical significance for ACC differences we used the methodology of Siegert et al. (2017), which also takes into account temporal autocorrelation by computing effective degrees of freedom.

In addition we further test the probabilistic forecast skill of the Best30 ensemble in comparison with the skill of the unconstrained ensemble with the ranked-probability skill score (RPSS; Wilks, 2011). The ranked probability score (RPS) is

computed by dividing each prediction made by the constrained ensemble (e.g. Best30) and the unconstrained ensembles





(here these predictions refer to the forecasts from each individual start dates) into three equiprobable categories (below normal, normal, and above normal), computing the terciles separately for observations and simulations to avoid the biases in mean and variance. The RPSS is then obtained by computing the relative difference between mean RPS of Best30 and the unconstrained ensemble (with positive values indicating the Best30 outperforms the unconstrained ensemble in terms of
probabilistic forecasts and vice versa). The same procedure was also applied for testing the added value of DCPP over the unconstrained ensemble in forecasting the first decade, i.e. years 1-10. The statistical significance of the RPSS is estimated by a random walk test following DelSole and Tippett (2016). For all skill evaluations the model fields of near-surface temperatures, sea-level pressure and precipitation, and the observational reference data, were regridded to a common 5°x5° grid.

**3 Results**

**3.1 Evaluation of the variability-constrained projections**

We evaluate the forecast quality of the constrained projections (where constraining decadal variability has the purpose to initialize decadal to multi-decadal predictions, similar to what is done in initialized climate predictions (e.g. Doblas-Reyes et al., 2013; Meehl et al., 2021) by means of data assimilation) by constructing hindcasts (also known as retrospective
forecasts) on three decadal and multi-decadal time ranges. For the first decade (average of forecast years 1-10, "FY1-10"), we also compare the skill of the Best30 ensemble with that of the actual DCPP ensemble obtained from the multi-model decadal predictions provided within CMIP6. We further evaluate the second decade ("FY11-20") and the 20-year forecasts ("FY1-20") in order to explore the applicability of the constraining approach beyond the ten year forecast period. Fig. 2(a-c) shows that the Best30 ensemble has high skill in terms of ACC for near-surface air temperature over most of the global
regions except in parts of the Pacific and southern ocean where the skill is statistically not significant (p>0.05). Similarly high positive ACC values are obtained for the unconstrained ensemble mean (not shown) suggesting that the global warming signal strongly contributes to these high correlations. To understand the additional skill of the Best30 over the unconstrained ensemble, residual correlations are shown in Fig. 2(d-f) after removing an estimate of the global warming signal from both the Best30 ensemble and the observations following the methodology of Smith et al. (2019). The same procedure is applied
to evaluate the skill of the DCPP ensemble over the unconstrained ensemble for FY1-10 (Fig. 2g). The results show that the Best30 residual correlations for the first decade are generally similar to DCPP in terms of the overall spatial distributions (cf. Fig. 2d and 2g). The Best30, however, shows larger added skill than DCPP in many regions including extended areas of the tropical Pacific, parts of Africa, eastern Asia, southern Europe and southeast Asia with residual correlations exceeding 0.6. In contrast, DCPP shows higher residual correlations than Best30 in the subpolar North Atlantic, which is a region where
previous studies have reported the largest added value in initialized decadal predictions (Doblas-Reyes et al., 2013; Yeager et al., 2018; Smith et al., 2019). Note that higher skill in the North Atlantic can also be achieved for the Best30 when constraining the projections using regional SST anomalies (see Section 3.3). Significant added value of the variability-




constraint is also found beyond the first ten forecast years typically covered by decadal predictions. We find positive residual correlations also for FY11-20 (Fig. 2e) and FY1-20 (Fig. 2d) over large parts of the Pacific, Atlantic and Indian Oceans and

some neighboring land regions including parts of Africa, Australia, eastern Asia and North America.

We further evaluate the efficacy of the constraining approach by means of the RPSS where positive values indicate superiority of the Best30 or DCPP over the unconstrained ensemble in making probabilistic forecasts (Fig. 2h-k). Similar to residual correlations, the Best30 shows significant added value over the unconstrained ensemble, and in several regions stronger added value than DCPP, for FY1-10, for example in the eastern tropical Pacific, the North Atlantic and Indian

Ocean, and neighboring land regions in most continents. Significant added value in terms of RPSS is also found for the longer forecast times, e.g. the second decade (FY11-20; Fig. 2i) and the next 20-years (FY1-20; Fig. 2j), over substantial parts of the global ocean and land regions. We note that the added value is found over similar regions across the different forecast times, which provides confidence in the robustness of the added skill from constraining decadal variability in the projections.

The constrained projections also show significant added value in predicting other variables than temperature, for example sea level pressure (Fig. 3). ACC values up to 0.8 and higher are obtained predominantly over the Pacific and the Atlantic oceans, parts of Northern Europe and Asia, the Southern Ocean and Antarctica (Fig. 3a-c). We again find added value in terms of ACC difference and RPSS over similar regions where also DCPP shows added value during FY1-10, mainly over the tropical Pacific and parts of the Atlantic ocean. Also for SLP, areas of added skill are found beyond the first decade

covered by DCPP, mostly over parts of the Pacific and Atlantic Oceans – indicating some SLP predictability on multi-decadal time scales.

The constrained projections also show some skill in predicting annual mean precipitation in land areas (Fig. 4). While we find significant skill in terms of ACC over large continental areas (e.g. Northern Eurasia, subtropical Africa, and South America, Fig. 4a-c), the added value for the Best30 compared to the unconstrained ensemble, as shown by their ACC

difference (Fig. 4d-f) is however generally small. Despite this lack of widespread added value, in some locations such as the Middle East, southern Africa, Australia, north and south America the ACC difference is positive and statistically significant at 95% confidence level for all three forecast periods. Note that also the DCPP initialized decadal predictions show only small added value for precipitation, and again there is some resemblance in the global patterns of added skill between Best30 and DCPP for FY1-10. Similarly, the Best30, as well as the DCPP, show limited added value over the unconstrained

ensemble also in terms of RPSS.

## 3.2 Sensitivity to different selection criteria

We next evaluate the sensitivity of the constrained projections to a number of choices related to the constraining criteria. There is a wide range of choices involved when applying the constraints, and it is beyond the scope of this paper to systematically document all possible choices and their effects. We rather aim to illustrate how different settings can be useful

to optimize the results depending on the targeted outcome. In particular we illustrate the sensitivity to (i) the temporal





averaging of SST anomalies used for constraining, (ii) the number of ensemble members kept in the constrained ensemble, and (iii) the metric based on which the 'best' members are selected.

Fig. 5 and supplementary Fig. S1 show residual correlation and RPSS results respectively when selecting the Best30 members using SST anomalies for different time periods instead of using 9-year mean SST anomalies. When selecting based on shorter time averages (e.g. 1 or 3 years), the added value of the constrained ensemble is smaller for decadal and multi-decadal predictions when compared to using longer averages (e.g. 9 years in Fig. 2). The overall spatial patterns of the residual correlations and RPSS are similar between the different selection periods, but values are lower and often not statistically significant when averaging over shorter periods. When using 6-year averages (Figures 5g-i and Fig. S1g-i), results are very similar to our default option of using 9-year averages. This suggests that low-frequency variability relevant for decadal to multi-decadal predictions is well constrained when using averages of 6 years or longer. However, the optimal choice for the averaging period depends on the particular prediction target. While averaging over six to nine years is suitable to constrain low-frequency variability and provides added value to predict the next decades, shorter time averages (filtering e.g. for inter-annual variability) can provide larger added value to predict just the next year, as illustrated in Fig. S2. Constraints based on 1-year averages lead to significant added value, measured in both residual correlation and RPSS, for forecast year 1 in the tropical Pacific, Indian Ocean, parts of Africa and Southeast Asia. In contrast, constraints based on averaging SST anomalies over 3 or more years yield almost no added value for forecast year 1. While significant added skill is found for forecast year 1 when constraining based on 1-year SST anomalies, the added skill is smaller than in the initialized DCPP predictions for this same forecast time.

Another choice is the number of ensemble members selected for the constrained ensemble. While for initialized climate predictions the benefit of using very large ensembles has been highlighted recently (Smith et al., 2020), the nature of our constraining method implies that simulations in less good agreement with the observed state would be included when selecting more members. In this context, the choice related to the number of selected ensemble members includes a balance between selecting only a few members more closely resembling the observed initial state or a larger constrained ensemble (which could more efficiently capture the predictable signal) that, however, also includes members with decreasing similarity of the initial SST anomaly patterns. The effects of this choice are illustrated in Fig. 6 and supplementary Fig. S3 , where we show the results for selecting the best 10, best 30 and best 50 members respectively. The results indicate overall a high robustness of the results to the number of selected members. All constrained sub-ensembles (of 10, 30 and 50 members) show very similar skill patterns, and added value in similar regions. The magnitude of the added skill (in particular for RPSS) is in some regions slightly larger for the smaller Best10 ensemble, however larger areas with significant added skill are found when using the Best50 ensemble (Fig. S4).

We finally test the use of a different metric to determine the level of agreement between the SST anomaly patterns in observations and the full set of CMIP6 ensemble members. Instead of calculating the pattern correlations, we use the area-weighted root mean squared error (RMSE), calculated based on the differences between observed and simulated SST anomalies over all grid cells (supplementary Fig. S5). Again we find broadly similar patterns of skill and added value




compared to the 'default' approach of constraining based on pattern correlations for 9-year average anomalies. This illustrates that using a globally aggregated error measure can also be useful to select the members in closest agreement with observed variability patterns. However, in our applications we find that the added skill with this alternative selection method is overall smaller than selecting based on pattern correlations. While we do not exclude the possibility that selecting based on RMSE can be advantageous for specific prediction targets, in our applications we find the selections based on pattern

correlations to yield higher skill.

### 3.3 Regional SST constraints and attribution of skill to specific ocean regions.

All results discussed so far were for constraints using global SST anomaly patterns, however constraining based on regional SST anomaly patterns can also be useful, either to optimize the skill over specific target regions or to understand the

predictive roles of certain ocean basins. Selecting the Best30 based on different SST regions can provide added value to regional scale projections of near-term climate. This is shown by constraining the Best30 ensemble using alternatively SST anomalies from the Pacific (65N-50S) basin or the North Atlantic (0-60N) basin (Fig. 7 and Fig. S6). These results show that Pacific constraints lead to substantially larger areas with significant added value than the Atlantic constraints (Fig. S7), with values that are similar to those obtained with the global constraints, which suggests that the Pacific ocean is a dominant

internal predictability source on decadal to multi-decadal timescales (compare Fig. 7, Fig. S6 and Fig. 2). Constraining based on North Atlantic SSTs, however, provides improved skill of the Best30 over mostly the sub-polar North Atlantic (Fig. 7 and Fig. S6 ) which is not seen when selecting based on either global or Pacific SSTs. Selecting based on Atlantic SSTs also provides some added value on multi-decadal time scales in parts of the Pacific, but overall the global areas with added skill over the unconstrained projections ensemble are smaller than when selecting based on Pacific or global SST anomalies (Fig.

S7). We demonstrate here the effects of constraining based on different ocean basins for the global picture of decadal to multi-decadal predictability. Using other more confined ocean regions, ideally physically informed, can thus be useful to optimize skill for specific locations or target regions (e.g. Borchert et al., 2021).

Selecting the "best" members based on regional SSTs can further be useful to attribute predictability to specific ocean regions, and thereby help generate understanding of the climate system. We illustrate this in the following for reproducing

the historical evolution of global average temperatures. Observed global average temperatures showed a slowdown in their increase rate during the early 2000s (Fig. 8), sometimes also termed as the 'hiatus' period (Easterling and Wehner, 2009; Cowtan and Way 2013; Trenberth, 2015; Fyfe et al., 2016). The HadCRUT4.6 time series shows a trend slope that is close to zero during 2003-2013, although the true global warming rate is thought to have been slightly larger when accounting for unsampled regions (Cowton and Way, 2013). However, no such warming slowdown is found in the ensemble mean of all

CMIP6 projections (which show an increase of 0.2K/decade during 2003-2013), indicating that forcing is unlikely to explain the reduced global warming rates during that time. The Best30 predictions 'initialized' in 1998 (i.e. constrained based on their SST anomaly patterns during 1989-1997) based on global SST patterns show a reduced warming rate of 0.13K/decade. And constraining based on Pacific SST yields an even smaller warming of about 0.10K/decade during 2003-2013,



confirming the important role of Pacific internal variability in modulating the 'hiatus' (Kosaka and Xie, 2013; England et al.,
2013). Our results further indicate that a reduced global warming rate during the one-and-a-half decades following 1998
would have been predictable based on the Pacific ocean temperatures in the preceding decade. No reduced warming rate is
found for the Best30 ensemble constrained based on North Atlantic SSTs, suggesting that the North Atlantic did not
contribute to this early 2000s global warming slowdown.

**4 Summary, discussion and conclusions**

We present a novel approach to constrain decadal-scale variability in large climate projection ensembles, acting as a 'poor-
man' initialization to align the phases of simulated and observed climate variability. The constraint selects those ensemble
members most closely resembling observed patterns of multi-annual SST anomalies. We apply this constraint to each year
from 1961 onwards to build a set of annually initialized hindcasts that cover multiple decades (i.e. as long as the projection
simulations are run). We evaluate the forecast quality of these constrained projections for the following 20 years after
applying the annual constraints, focusing the evaluation on the average of forecast years 1-10 (i.e. the forecast period also
covered by initialized decadal hindcasts e.g. from DCPP), 11-20, and 1-20. For all these forecast times, the constrained
ensemble provides skillful predictions of near-surface temperature, sea-level pressure and precipitation in large areas of the
globe. Significant improvements over the unconstrained ensemble are found in particular for near-surface temperature and
sea-level pressure.

The skill of the variability-constrained projections for predicting the first decade is comparable to the skill provided by the
DCPP decadal hindcasts. In particular for near-surface temperature, the constrained ensemble provides added skill over the
unconstrained large ensemble of projections in larger global areas than DCPP, in particular in the Pacific, where initialized
decadal prediction systems tend to have problems (see e.g. Yeager et al., 2018). This indicates that there is decadal-scale
predictability in the climate system that is missed by current initialized decadal prediction systems which typically suffer
from initialisation-related effects perturbing the model attractors, such as shocks and drifts (see e.g. Bilbao et al., 2021). The
'poor-man' initialisation, by selecting well span-up projection members that are in phase with observed variability, does not
involve such perturbation of the model attractor, and therefore does not introduce such artifacts. Furthermore, while
initialized decadal predictions require large computational resources, the 'poor-man' initialization presented here makes use
of existing climate projections and does not require to run any additional simulations. Constraining against observed SSTs
also allows to produce hindcast sets covering much longer time periods (as far back as suitable SST observations are
available to make the constraints). Here we start the hindcasts in 1961 for comparability with DCPP. It would also be
relatively straight-forward to use the presented approach to provide predictions in near-real time
(https://hadleyserver.metoffice.gov.uk/wmolc/). Such predictions can be done as soon as observational SST fields are
available and can also be used as a benchmark for operational decadal prediction.

While initialized climate predictions such as those in DCPP are typically restricted to predictions of the next 10 years after
initialization, the predictions based on constrained projections can easily (i.e. at no extra cost) provide climate information





beyond 10 years. We identify significant added value from the variability-constraint also in the second predicted decade (e.g. forecast years 11-20), and when predicting multi-decadal averages (e.g. forecast years 1-20). These results indicate that there
is significant multi-decadal predictability from internal climate variability, which can be exploited to improve near-term climate change estimates.

In this study we discuss some sensitivity of the results to different choices when implementing the constraint, in particular to the averaging time, to the size of the constrained ensemble, to the ocean regions used to evaluate the agreement between models and observations, and to the metric to quantify agreement. These choices can lead to increased or decreased skill for
specific regions and prediction time scales. The sensitivity tests therefore also indicate the possibility to optimize the skill for specific applications, e.g. finding the settings that lead to the highest forecast quality for a specific forecast time at a specific location.

We further demonstrate that constraining variability in climate projections can be useful to attribute predictability to certain ocean regions, and thus help generate understanding of the climate system. By applying the constraint only to certain ocean
regions, we can evaluate the regionally constrained ensembles in their ability to predict certain climate phenomena. On the example of the so-called 'hiatus' in global mean temperature increases in the early 2000s, we demonstrate that the projections constrained to observed climate anomalies leading up to 1998 are capable of predicting a slowdown in global warming rates during the following 15 years. This predictability can be attributed to constraining Pacific variability (in agreement with previous studies based on specific model experiments prescribing aspects of Pacific variability, such as
Kosaka and Xie (2013) and England et al. (2013)).

Analogue-based selections of climate simulations have been proven useful in the context of both seasonal and regional decadal predictions (Ding et al., 2018; Menary et al., 2021). Using SST and sea surface height anomalies, and selecting those simulations with minimum distance to the target climate states, Ding et al. (2018) demonstrated skill of such sub-selected analogues in predicting observed monthly mean SST and sea surface height anomalies mainly in the tropical Indo-Pacific
region over the following 12 months. Menary et al. (2021) used a pool of 35-year mean SST anomalies from different CMIP5 and CMIP6 experiments to find model analogues with the lowest error in spatial patterns compared to observed states of 35-year mean SST anomalies, to make predictions for years 2-10 in the North Atlantic region.

The constraining approach used here is similar to Mahmood et al. (2021), who constrained a 40-member single-model ensemble based on the pattern agreement with initialized decadal predictions. The major differences are that here we select
members from a much larger multi-model ensemble, and we select based on the agreement with SST anomaly patterns from observations instead of decadal predictions. The benefit from the constraint (in terms of added skill) found here is substantially larger than reported by Mahmood et al. (2021), likely due to the large ensemble size of CMIP6 projections to select from. Also, constraining based on decadal predictions requires that these decadal predictions add skill over the projections, which only happens in certain regions like the North Atlantic (Befort et al. 2021). Limitations that deteriorate
the skill in decadal predictions (e.g. Bilbao et al., 2021) may however transfer to the constrained projections. Future work



will test the decadal predictions based constraint on the large multi-model ensemble of projections as used here, to understand if the larger ensemble size also benefits that approach.

Both approaches, constraining the projections based on their agreement with either observations or or decadal predictions, can be used to provide seamless climate information for the next multiple decades. This is an important advantage over the

use of different datasets for different time scales, e.g. initialized seasonal to decadal predictions for the first few years and projections afterwards. Data from these different sources are often inconsistent, both in a statistical sense and the climate conditions that they represent. In contrast, the variability-constrained projections provide consistent transient climate information for the next years and multiple decades, which can facilitate their use when seamless climate information across timescales is required. We show here that such constrained projections show promising skill, comparable to initialized

predictions on the decadal time scale, and provide significant added value over unconstrained projections on multi-decadal time scales - pointing to predictability in the climate system that is not currently exploited with existing prediction systems. The variability-constrained projections therefore provide a promising pathway to provide improved climate information, of reduced uncertainty and increased accuracy, about near-term climate change in the next few decades. This improved information can be important to underpin targeted adaptation strategies.


**Acknowledgements**

This research was funded by the Horizon 2020 EUCP EUropean Climate Prediction system (grant number 776613). MGD and PO are grateful for funding by the Spanish Ministry for the Economy, Industry and Competitiveness grant references RYC-2017-22964 and RYC-2017-22772, respectively.We acknowledge the World Climate Research Programme, which,

through its Working Group on Coupled Modelling, coordinated and promoted CMIP6. We thank the climate modeling groups for producing and making available their model output, the Earth System Grid Federation (ESGF) for archiving the data and providing access, and the multiple funding agencies who support CMIP6 and ESGF.

**Author contributions**

MGD, PO, FJDR and RM designed the study. RM performed the analysis with active guidance from MGD, PO, and FJDR. RM and MGD drafted the manuscript. CDT helped with the probabilistic forecast quality assessment. MS and PAB downloaded and managed the large datasets of CMIP6 simulations and the observations. All authors contributed to the writing.

**Competing Interests**

Authors declare that they have no competing interests.



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
