# Peer review of "Constraining low-frequency variability in climate projections to predict climate on decadal to multi-decadal time scales - a 'poorman' initialized prediction system"

_EGUsphere, 2022_

## Author Response (AR1)

**Reviewer 1:**

In my opinion this is a very well written and clear paper which presents an interesting new way of combining a large multi-model dataset in order to constrain future projections. As such I have very few comments to make and I believe that this paper should be published with only a few corrections.

The results are impressive and warrant publication. However the one area I would like to see more explanation on is the cause of the skill found. In the summary (line 300) you claim that "these results indicate that there is significant multi-decadal predictability from internal variability". I am not convinced though that as they stand your results can justify this claim (although I'm not suggesting this is necessarily incorrect). By selecting the best 30 (or similar) simulations from the CMIP6 archive based on the 9 most recent years, could it not be possible that what you are doing is weighting your results to certain models (which is in itself a rich area of literature). This could be those models with the most realistic response to forcing, especially when outside the anomaly period at the start or end of the record. Or it could potentially be those with the most realistic modes of variability. Both of which, I would have thought could theoretically give an increase in skill over the next decades. This is especially pertinent, given that you state that the method gives a better constraint that a single model ensemble (although as you say this is expected given the increased number of simulations to sample from). I think that this point needs to be discussed, and preferably investigated. One simple test you could do, without the need for further analysis, would be to check if you are selecting simulations preferentially from one climate model or if the 30 members are selected from the full model range. In addition it would also be interesting to see if the skill varies through time.

Author Response:

We thank the reviewer for the encouragement and the constructive insights. The reviewer is right that the added skill can potentially also be due to selecting those models with a more realistic response to forcing, or representation of variability modes. We have extended our analysis to provide some insight into characteristics of the variations that may be related to the enhanced skill in the constrained ensemble, and added/adjusted some discussion in this regard.

First, we added some time series plots for the regions where the temperature hindcasts of the constrained ensemble show added skill over the full CMIP6 ensemble (e.g. tropical Pacific, north Atlantic and eastern Asia (supplementary Figure S2)). These figures indicate improved long-term behavior of the constrained ensemble, which may be related to the representation of trends but also multi-decadal variability. In particular, the constrained ensemble better captures the cold anomalies in these regions in the early part of the hindcast where both the unconstrained CMIP6 and DCPP ensembles are warmer than observations and the Best30 constrained ensemble. The constrained ensemble also better captures observed decadal-scale variations in the warming rate in these regions, whereas the warming rate in the all-member CMIP6 ensemble is more homogeneous in time. We have added some discussion on these characteristics (lines 325-331): "*The added skill in the constrained projections likely comes from an improved representation of long-term changes in response to forcing (as also found for decadal predictions, e.g. Doblas-Reyes et al., 2013), and also the representation of decadal-scale variations. Inspection of regional average time series in regions with added skill (e.g. in the Pacific, eastern Asia or the North Atlantic) indicates warming trends more similar to the observations in the constrained ensemble compared to the full CMIP6 ensemble in particular in the early parts of the hindcast period. These time series also show that the constrained ensemble better captures the observed variations around the warming trend, likely in relation to*

*decadal-scale climate variability*".

In addition, we also checked statistics on how often the different models are selected, and this can in fact be fairly uneven for some start dates (but note that also in the full, unconstrained, ensemble there are substantial differences about the number of runs contributed by different models). Figure R1 (below) shows that the CanESM5 model (which provides 25 ensemble members) is selected most frequently for most start dates, and also e.g. MIROC6 (providing 50 members) and MIROC-ES2L (providing 30 members) are chosen more frequently than other models, however the selection frequency is overall not proportional to the number of ensemble members provided per model.

To further test if the skill is due to the over-proportional selection of some models, we also implemented the constraining method so it limits the number of runs that can be selected from any models to 5 or 3 members (see Figure R2). The pattern of skill for these selections is mostly similar to the results shown in the main text (although slightly lower), which suggests that the role of overly selecting just a few models may be rather limited, and points to an important part of the skill we find being indeed due to the phasing of internal variability. These two figures are now added in the supplementary materials as Figures S12 and S13.

Finally, the question about temporal variations of skill is indeed an interesting one. As we are focusing on decadal to multi-decadal time scales here, the degrees of freedom of the time series are already relatively low. It would be interesting to address this question in the future, as the 'initialisation' in comparison to observed SST patterns would in theory allow us to extend the set of hindcasts substantially back in time. However, in this study we prefer to start the hindcasts in 1961 to allow comparison with the DCPP-A decadal hindcasts.

[Figure]

**Figure R1:** Count of how often each model is selected as part of the Best30 ensemble for each start date. The total number of ensemble members used are shown along with model names. The circle size indicates the count of how many ensemble members from each model are selected.

[Figure]

**Figure R2:** As Figure 2 of the manuscript, but limiting the number of members for a model to be selected as part of Best30 to a maximum of 5.

I am also somewhat surprised that you have not shown results for future projections, since that would seems to be the logical direction for this type of analysis. Although, perhaps this is being left for future work (which would be perfectly justifiable).

Author Response: We would like to understand some features of future projections in more detail, in particular as it also involves effects related to the different climate sensitivities of the models, and plan to present this in dedicated future work.

More minor points:

L 84. I think that adding the acronym for the Extended Reconstructed Sea Surface Temperature Version 5 dataset would be helpful.

Author Response: Thank you, we added in the revised version of the manuscript the acronym ERSSTv5.

L88. Why was a reference period of 1981-2010 chosen, and are the results sensitive to this?

Author Response: That chosen reference period lays roughly in the middle of the investigation period. However, as we are using anomalies, the choice of reference period would only affect the magnitude and sign of the local anomaly values, but not the patterns of where the high and the low values are (which is what is important for the constraint based on pattern correlations). We also repeated the analysis with the longer 1961-2010 reference period, and as expected the results regarding ACC and residual correlations are very similar to the results shown in Figure 2 (see Figure R3 below):

[Figure]

**Figure R3**: Residual correlations (a-c) and RPSS (d-f) for the Best30 ensemble means similar to the results shown in Figure 2 of the manuscript but based on anomalies computed with a reference climatology of 1961-2010 period.

L117. Was the forced signal that you removed the multi-model mean?

Author Response: Yes, we now clarified in the text that the forced signal was estimated from the ensemble mean of all 212 CMIP6 members.

L151. Why global warming and not external forcing in general? I would have thought that anthropogenic aerosols and volcanic eruptions might also have an impact.

Author Response: You are right, we have replaced 'global warming' by 'external forcing'. Thank you.

L261. It would be useful to cite papers which have suggested that forcing (particularly natural) could contribute to the slow down, (see e.g. box 3.1 IPCC AR6 WG1)

Author Response: Thank you. We have added a brief note also mentioning studies that highlighted forcing (e.g. related to small volcanic eruptions) contributing to the slowdown in global warming during that period.

**Reviewer 2:**

The paper presents an interesting approach to providing climate predictions based on constraining non-initialised climate projections with observed climate variability. Only those ensemble members of climate projections that show the largest agreement with observed SST anomalies in years prior to the forecast start date are used to construct climate predictions. Instead of a full initialisation with the observed state as normally done in climate predictions (e.g. for seasonal and decadal prediction), a simplified ('poor-man') approach of aligning the phase of theses of simulated and observed SST variability is used. After applying the approach to hindcasts from 1961 onwards, the forecast quality of the predictions is evaluated and compared to both fully initialised decadal predictions and unconstraint climate projections. It was concluded that the constrained ensemble provides skilful predictions of near-surface temperature, lea-level pressure and precipitation in large areas of the globe. During the first decade of predictions the skill of the poor-man predictions is comparable to the initialised decadal predictions. Significant added value from the constrained approach was found in the second decade for which initialised predictions are not available. Sensitivities to certain choices like the past period and geographical regions of the constraint, ensemble size and skill metric, have been discussed.

I think the approach explored in this study is very interesting and certainly deserves to be published. In particular, I agree with the authors that the potential benefits of their approach over both initialised decadal predictions and unconstrained projections for providing seamless climate information could be big and important. However, I cannot recommend publication of the paper in its current form because it lacks several critical aspects that are discussed below.

Author Response: We thank the reviewer for the constructive comments that helped us to further clarify some aspects of the manuscript.

**Major comment**

In my view the manuscript suffers substantially from the poor demonstration of the results. While the motivation and the methodological approach are nicely laid out, the analysis of the results and their graphical presentation do not provide enough evidence to the reader to be convinced of the benefits that the new prediction system might bring. With "enough evidence" I don't mean the quantity of analysis or plots but rather the opposite: the authors have taken the approach to include into the manuscript and the supplementary material almost every possible plot one can think of for the quantities they have analysed. However and unfortunately, the large number of plots does not provide an equally large amount of useful information. I would suggest to critically review all plots and only show those which really help support the claims you are trying to make. It is the responsibility of the authors to make a meaningful selection of the diagnostics that help tell the story you wish to convey and should go into a publication. This critical selection should not be left to the reader alone. I have the following specific recommendations:

Author Response: We thank the reviewer for pointing out this shortcoming that made it difficult to follow the clear line of the story in our submitted manuscript. In our revision we have critically considered the relevance of the different plots, and moved in particular those map plots related to sensitivity tests to the supplementary information. We have in their place included a new plot (new Figure 5) that summarizes key information of those sensitivity tests. Now the main text

figures are reduced to those more relevant to presenting the constraining method, and we hope the reviewer agrees that this improves the readability of the paper.

Fig 1: I think this could be cut short without loss of information by only showing one start date as a demonstrator and carefully describing the methodology in the text and figure caption, as already done.

Author Response: We have carefully considered the suggestion by the reviewer, but feel that it is useful to keep showing several start dates to also illustrate the point that the selection differs for each (start) year. When presenting the work on different occasions we have noticed some audiences misunderstood the method by missing the important information that another sub-ensemble is selected in each (start) year. We therefore think that showing several start dates in the figure illustrates the method more completely, despite making the figure a little more complex.

Fig 2:

- I don't find showing means over 10 or 20 years are helpful in the prediction context. The window is too long to provide useful information. It would be better to split the windows into smaller ones to identify those time ranges where the approach can improve either decadal predictions of projections. For example, if the added value over non-initialised projections kicks in after 10 years, it would be most interesting to know when this happens – it is just immediately after the 10 years or more towards the end of the 20-year period? Averages over 10 years smear out the impact, and means over 20 years can potentially even be misleading by implying the skill comes from the later years when most likely it is coming from the earlier years. I would suggest looking at 1-5, 6-10, 11-15 and 16-20 years. Or, if the results reveal interesting insight, even for finer forecast ranges. This recommendation applies to almost all plots in the paper and supplementary material.

Author Response: Please note that our constraining approach presented here is targeted at aligning low-frequency variability between the simulations and the observations, with the explicit goal to refine near-term climate information on decadal to multi-decadal time scales. We therefore prefer to keep the presentation of results with a focus on the different 10-year and 20-year periods, which are representative for near-term climate change as opposed to predicting inter-annual variations.

Following the suggestion by the reviewer, however, we have also added a new Supplementary Figure S3 which shows the results for the different 5-year periods of years 1-5, 6-10, 11-15 and 16-20. The skill patterns for these shorter periods are largely consistent with those of the decadal and multi-decadal averages. We have added a paragraph to discuss the skill for these pentadal forecast periods.

- Please also show ACC of the unconstrained projections after 10 years to provide a reference to which to compare to. Fig 3 for SLP shows differences which is helpful but Fig 2 for surface temperature does not.

Author Response: We added a Supplementary Figure to show ACC of the unconstrained projections (new Figure S1).

- It would also be interesting to show how a similarly constrained decadal prediction

ensemble would perform, that is sub-sampling those ensemble members from DCPP that most closely resemble the past SST observation after e.g. forecast year 1. That would of course imply that the predictions are only useable after applying the constrain (e.g. after 1 year) but for the longer time scales this could still be useful.

Author Response: We agree that applying similar constraints also to the initialised decadal predictions is a very interesting and promising perspective. However, we don't see how this can be implemented in a consistent way to constraining the projections in this same paper (using different selection periods, which would introduce inconsistencies), and therefore suggest that such constraint of decadal predictions based on their agreement with (early) observations should be the topic of future research.

- What is the reference forecast used in the RPSS computation? Please add this information in the figure caption.

Author Response: The reference forecast for RPSS is the full, unconstrained, CMIP6 ensemble, which has been mentioned explicitly in the figure caption.

Since showing too many global maps is not sustainable, I would recommend to condense the critical information either into 2D plots or bar charts (similar to what has already been done in the Supplement but for finer forecast ranges). These could be good options for the various sensitivity studies. For example, global or key regional scores could be plotted in a 2D plot as a function of forecast year and selection period to replace Fig S2 etc. Such a condensation would make space to show a direct comparison (or differences) with the performance of the unconstrained ensemble or the decadal predictions.

Author Response: Thanks for pointing this out, and we agree with your criticism. In the revised manuscript we now have moved all map plots related to the sensitivity tests (i.e. previous Figures 5, 6, 7) to the Supplementary information, and summarise some global information from these figures in a new (bar plot) figure (new Figure. 5), as suggested by the reviewer.

I find some of the results are a bit over-interpreted and should be re-worded slightly more carefully. For example, on line 173 you say that added value is found over similar regions across different forecast times providing confidence in the robustness. However, the plots these lines refer to (Fig 2h-j) indicate for example some inconsistencies in the North Atlantic and the tropical East Pacific. Or for SLP in Fig 3, the highly skilful subtropical North Atlantic for FY11-20 (Fig 3b) is not showing during the first 10 years (Fig 3a). Why is this? Around lines 180, mention the problematic issues over the Indian Ocean.

Author Response: In our revision we reworded the sentence in line 173 that claimed robustness from similar regions with added value across different forecast times as follows: *"While many regions consistently exhibit added value from the constraint for the different forecast times shown, in other regions such as large parts of the Atlantic Ocean or the tropical Indian Ocean positive residual correlations emerge only in the second decade of the hindcasts."*

We also added a sentence mentioning the issues over the Indian Ocean after line 180 (see new Lines 201-202): "*Some added skill also emerges only in the second forecast decade e.g. over parts of the subtropical Atlantic and the Indian Ocean (noting however that ACC over the Indian Ocean remains negative for all forecast periods shown).*"

The result that the constrained projections can outperform the initialised predictions is very

interesting. I feel it would require some more discussion as to what the mechanisms are that can lead to this perhaps surprising skill. Discussing potential explanations would make the paper much stronger than simply describing it.

Author Response: We agree that it is intriguing to understand what leads to the higher skill in the constrained projections, as also noted by reviewer 1. Inspection of regional time series in regions with added skill shows that the constrained ensemble shows improved representation of both long-term trends and multi-decadal variability. This may be partly related to the constrained ensemble better representing the forced climate response, but also to a preferred selection of some models with a better representation of either long-term changes and variability. Another important point is that the constrained ensemble members all represent undisturbed model attractors under transient forcing, whereas the initialised decadal predictions suffer from initialisation shocks and the drifts inherent to initialization. We have added some discussion throughout the text, but think a deeper investigation of processes is beyond the scope of this paper as it would require retrieving a range of different variables from the >200 ensemble members.

*Supplementary Information:*

It is not clear which variables have been analysed in Fig S1-S3 and S5-S6.

Author Response: We specified in the captions of Figs S1-S3 and S6 that the results are for near-surface temperature. Fig. S5 specified "same as Fig. 2".

**Minor comments:**

Fig 4 caption: unclear what exactly is meant by added skill – please specify.

Author Response: We understand this comment refers to Supplementary Figure S4 (previous number) and the results from that figure are now part of new Figure 5 in which we have clarified in the caption that added skill was "measured as residual correlation and RPSS against the unconstrained CMIP6 ensemble as reference"

Switching between the ACC and the residual correlations introduces some inconsistencies in the manuscript. For the purpose of this paper, it might be sufficient to only show ACC. Fig 5 and 6 could go in the Supplement.

Author Response: To avoid this confusion of using different measures to identify added skill, we now show residual correlations for all variables (Figures 2, 3, 4).

Figures 5,6,7 are now moved to supplementary material, and their key conclusions summarised in the (new) Figure 5.

Why are the atmospheric fields computed on a 5x5 degree grid and not on a 3x3 degree grid as the SSTs?

Author Response: We followed recommendations for decadal predictions (Goddard et al., 2013), with the rationale that remapping atmospheric fields to a coarser grid removed small-scale noise in the skill analysis. For the constraint based on SST data we give preference to a

slightly finer common grid (within what is possible given some models still have a relatively low resolution), to also capture effects of finer scale variability patterns.

Section 3.1: The text could be improved by introducing more paragraphs and reducing the use of brackets ().

Author Response: Thanks for pointing this out. We have slightly rewritten parts of Section 3.1, to split up overly long paragraphs and reduced brackets, to improve readability.

Sensitivity to temporal averaging of SST anomalies, around lines 206-209: emphasise this finding more as very interesting.

Author Response: We added a sentence to emphasise this point of shorter averaging periods being favourable for inter-annual predictions, (see lines 232-235): "*This is plausible as shorter averaging periods will emphasise the signals related to inter-annual variability in the member selections, whereas longer averaging periods will emphasise signals related to lower frequency variability relevant for predicting variations on decadal time scales.*"

The cited reference of Menary et al (2021) sounds very relevant – could you expand on your discussion of this paper in Section 4?

Author Response: We have expanded the discussion of the Menary (2021) reference, which we agree is very relevant in the context of our study.

Sensitivity to ensemble members: Why have you stopped sub-sampling at 50 members? It would be interesting to see the convergence for the full ensemble. Would it be possible to show a plot showing this convergence for perhaps a global quantity? Is there an optimal ensemble size?

Author Response: As discussed in the paper, a larger ensemble size of sub-selected projection members means to include members that are less similar at initialisation, and therefore may deteriorate the skill. We therefore do not expect a "convergence" of the skill as in e.g. in DCPP initialised decadal predictions. The optimal ensemble size (similar as the other sensitivities we discuss) will depend on the region and variable of interest. We tried to highlight this in our discussion of the different sensitivities.

Figure 7: show comparison with global pattern (repeat from Fig 1). Also show comparison with unconstrained and decadal prediction. Makes interpretation of these results easier.

Author Response: The maps in (previous) Figure 7 (now supplementary figure S4) are directly comparable to the maps based on global constraints in Figure 2. To avoid redundancy of results between figures we prefer not to repeat the "default" setting results for all of the figures showing sensitivity tests. However, in the revised manuscript we now included a new overview figure (new Figure 5), which compares the global areas of added skill for all settings, and this figure also includes the bars for the "default" shown in Figure 2.

Fig 8: Can you speculate as to why the constraint over the North Atlantic makes the forecasts worse? Fig 8b is not needed.

Author Response: We also checked the skill/added skill for GMST over the entire hindcast period (i.e. not just the hiatus period shown here), and e.g. the residual correlation based on the

North Atlantic constraint is negative, meaning this constraint does not improve the GMST predictions over the full ensemble (whereas residual correlations for the global and Pacific constraint are positive). We prefer not to speculate about reasons in the manuscript, but added the information about the lack of added skill and that this suggests the North Atlantic not controlling variations in GMST (lines 295-300):

*"No reduced warming rate is found for the Best30 ensemble constrained based on North Atlantic SSTs, suggesting that the North Atlantic did not contribute to this early 2000s global warming slowdown. Note that, also considering the entire hindcast period, the North Atlantic constraint does not improve GMST predictions compared to the full unconstrained ensemble (i.e. residual correlations are negative, not shown). This suggests that, at least based on the models used, the North Atlantic does not seem to provide predictability for global mean temperature.*

Fig 8b has been removed, and we included the slope values it previously showed in the legend.

*Supplementary Information:*

Figures S8 – S10: instead of repeating maps very similar to Fig 2—4, it would perhaps be more informative to show differences to these other maps.

Author Response: The purpose of these figures is to show that the main conclusions hold regardless of reference dataset used. We think that this message is better delivered by repeating the figures, as this allows us to confirm that the broad global patterns and features are robust. A difference map might be more useful to measure the magnitude of the uncertainties, but this was not the intended purpose of this analysis.

**Reviewer 3**

**General Comments:**

This manuscript by Mahmood et al. (egusphere-2022-98) explores an alternate way of decadal to multidecadal predictions by subsampling individual CMIP6 historical simulations that better matches observed SST at a given time (start dates) and by tracking the trajectories of the subsampled simulations for the next two decades (analogue method hereafter). The authors show that the added value of the analogue method over the uninitialized simulations is comparable to that of initialized decadal prediction simulation (DCPP). The manuscript is overall well organized and written. I have enjoyed reading the manuscript. I also acknowledge the thorough effort to test the sensitivity of the results to subsampling criteria. I think the manuscript has potential to draw attention from climate research community, as the proposed method can leverage existing simulations in the application of the prediction of climate, instead of running computationally expensive decadal prediction simulations. However, before I recommend accepting for publication, I have a couple of specific comments that I wish the authors further demonstrate or explain along with some minor suggestions.

Author Response: We thank the reviewer for providing these supportive and constructive comments, which we believe have helped to clarify some important aspects of our study.

**Specific Comments:**

1) The authors shows that the analogue method exhibits high skill in the Pacific Ocean, even higher than skill in the subpolar North Atlantic, which even lasts for FY11-20 . Such a long memory in the Pacific is surprising and in stark contrast to the current understanding that predictability in the Pacific Ocean is low on decadal timescales while very high in the subpolar North Atlantic. The low predictability in initialized decadal predictions may be related to initialization shock/drift, as the authors also discuss in the manuscript. However, the low predictability in the Pacific Ocean is also pervasive in "perfect model" experiments (e.g., Collins 2002; Pohlmann et al. 2004), which does not suffer from initialization shock/drift. Why the author's analogue method shows such superior skill in the Pacific Ocean? Isn't this high skill possibly related to the forced signal that is not completely removed by the method the authors used (Smith et al. 2019)? One way to verify this is to perform a bootstrapping method for the statistical test, rather than Student's t-test. If the ACC from Best30 is found outside of the (eg., 2.5 to 97.5 percentile) distribution of the ACCs from randomly sampled 30 members, assuming that the uninitialized ensemble mean used in Smith et al.'s method is the total 212-member ensemble mean, the authors can say more confidently that the high Pacific skill is indeed not from the forced signal.

Author Response: We thank the reviewer for this comment, which is also echoed by some comments from the other reviewers.

We have implemented the bootstrapping method as suggested by the reviewer, and find the ACC of the Best30 ensemble in the Pacific is outside of the 2.5 to 97.5 percentile distribution of randomly sampled 30 members (Figure R4). In fact the areas with significant skill or significant skill differences are very similar compared to Figure 2 in the manuscript, suggesting our results are robust and not just an artifact from incomplete removal of the forced signal.

In addition, we have also compared global trend maps and regional time series, and these give

some indication that the long-term changes in Best30 are (slightly) more similar to observations than long-term changes in the full ensemble in areas that exhibit added skill. This suggests that the added skill may, at least in part, also arise from a more realistic estimate of long-term changes/trends as a consequence of initialisation. We have extended the discussion to also cover this aspect and added a new supplementary Figure S2. The added text reads as (lines 325-331): "*The added skill in the constrained projections likely comes from an improved representation of long-term changes in response to forcing (as also found for decadal predictions, e.g. Doblas-Reyes et al., 2013), and also the representation of decadal-scale variations. Inspection of regional average time series in regions with added skill (e.g. in the Pacific, eastern Asia or the North Atlantic) indicates warming trends more similar to the observations in the constrained ensemble compared to the full CMIP6 ensemble in particular in the early parts of the hindcast period. These time series also show that the constrained ensemble better captures the observed variations around the warming trend, likely in relation to decadal-scale climate variability.*"

[Figure]

**Figure R4**: ACC (a-c), residual correlation (d-f) and RPSS (g-i) that are the same as shown Figure 2 in the manuscript but the stipplings here represents values that lie within 2.5th and 97.5th percentile range of the corresponding 1000 distributions obtained by by a bootstrapping method randomly selecting 30 members at each start date.

2) If that is the case, why the skill is so high in the Pacific? Since this would the most important finding of the study, in my opinion, as it is in contrast to the current understanding, I recommend that the authors further demonstrate the reasons for the high Pacific skill.

Author Response: Please see our response to the previous comment. A comparison of long-term changes e.g. in these Pacific regions indicates that the constrained ensemble seems to show more realistic trends in particular during the earlier part of the hindcast period, but also improved representation of decadal-scale variabiliy. These different aspects are now discussed in the text and we have added a new supplementary Figure S2.

3) The Atlantic skill is low for FY1-10, but picks up for FY11-20 (Fig. 2d-e). Why is this the case? I think the low skill for FY1-10 is because Best30 is dominated by the correlations in the Pacific and as demonstrated in the regional SST constraints, but it is hard to understand why there is an rebound in ACC skill.

Author Response: This is an interesting feature, which we are not able to fully explain at this point. Please note that in the analysis of pentadal forecast periods (new supplementary Figure S3) we do find added skill in the Northeast Atlantic during years 1-5 (similar to DCPP), which disappears however in the second pentad.

4) The authors introduce several statistical methods in section 2, without a description, just by referring to citations. I recommend adding a brief description for each method.

Author Response: We have added brief explanations for the different methods in Section 2.

**Technical corrections:**

l. 43: Remove "in" after phasing.

Author Response: We have reworded "phasing in" with "aligning the phases of".

l. 88: …anomalies "relative to" the reference climatological period…

Author Response: Thank you, we reworded as suggested.

---

## Author Response (AR3)

**Comments to the author**:

Dear Authors,

Thank you for the latest set of updates to your manuscript. I have some additional minor suggestions for improving the graphical presentation of the results, before accepting your study for publication in Earth System Dynamics.

**Author response:**

We thank the editor for providing these additional suggestions to improve the overall presentation of our work. We have incorporated all your suggestions in the revised documents and provide responses to each specific suggestion in the following.

Thank you also for efficiently handling the review of our manuscript.

- Fig. 5 You could add labels "residual correlation" and "RPSS" to the two panels

**Author response:**

Done. Thank you.

- Supplement: Text S1 appears in the middle of the document, please move to the top.

**Author response:**

We moved the Table S1 at the top of the supplementary materials so that the text S1 is exclusively followed by the relevant figures discussed in this text, i.e. the structure of the Supplement is now: (i) Supplementary Table, (ii) Supplementary Figures quoted in the manuscript text, (iii) Supplementary text to discuss observations-related uncertainties, including the figures relevant for this supplementary text.

- Fig. S3 I understand that you tried to arrange this is the same format as the other figures, but having four maps per row makes the resulting figure very small and difficult to read without needing to zoom in on the single panels. Is there the possibility of changing the layout to have maximum three maps per row (e.g. by arranging them vertically, as you did in Fig. S6)?

**Author response:**

The Figure S3 is now split into two Figures (one for Residual Correlation and one for RPSS) with panels arranged vertically as suggested. We keep the comparisons between the Best30 and DCPP in a row for each forecast period to easily facilitate the direct comparisons between the two predictions for the common forecast periods.

As for further reducing the number of SI figures/removing some of the sensitivity tests, I leave this to your discretion.

**Author response:**

We are grateful to the editor for supporting the option of keeping/removing the SI figures. We prefer keeping the figures showing RPSS as well. As outlined in our previous response, we believe

the different skill metrics are complementary, and some readers may be interested in seeing the spatial details of the results summarized in the main text figure 5.